# Magnetically induced currents and aromaticity in ligand-stabilized Au and AuPt superatoms

Omar López-Estrada [1], Bernardo Zuniga-Gutierrez [2], Elli Selenius [1], Sami Malola[1] & Hannu Häkkinen [1,3]✉

Understanding magnetically induced currents (MICs) in aromatic or metallic nanostructures is crucial for interpreting local magnetic shielding and NMR data. Direct measurements of the induced currents have been successful only in a few planar molecules but their indirect effects are seen in NMR shifts of probe nuclei. Here, we have implemented a numerically efficient method to calculate gauge-including MICs in the formalism of auxiliary density functional theory. We analyze the currents in two experimentally synthesized gold-based, hydrogen-containing ligand-stabilized nanoclusters $[HAu_9(PPh_3)_8]^{2+}$ and $[PtHAu_8(PPh_3)_8]^+$. Both clusters have a similar octet configuration of Au(6s)-derived delocalized "superatomic" electrons. Surprisingly, Pt-doping in gold increases the diatropic response of the superatomic electrons to an external magnetic field and enhances the aromaticity of $[PtHAu_8(PPh_3)_8]^+$. This is manifested by a stronger shielding of the hydrogen proton in the metal core of the cluster as compared to $[HAu_9(PPh_3)_8]^{2+}$, causing a significant upfield shift in agreement with experimental proton NMR data measured for these two clusters. Our method allows the determination of local magnetic shielding properties for any component in large 3D nanostructures, opening the door for detailed interpretation of complex NMR spectra.

[1] Department of Physics, Nanoscience Center, University of Jyväskylä, Jyväskylä, Finland. [2] Departamento de Química, Universidad de Guadalajara, CUCEI, Guadalajara, Jalisco, Mexico. [3] Department of Chemistry, Nanoscience Center, University of Jyväskylä, Jyväskylä, Finland. ✉email: hannu.j.hakkinen@jyu.fi

Magnetically induced currents (MICs) originate from the response of electrons of an object placed in an external magnetic field B[1,2]. A diamagnetic object (atom, molecule, cluster, nanoparticle, nanostructure) generates an internal ring electron current in a plane normal to B in such a way that it induces a magnetic field opposite to the external one. This classical ring current is called a diatropic current, increasing the shielding of the part of the object inside the current ring against the effect of the external field. Systems with strong diatropic currents are called aromatic[2,3]. Antiaromatic systems have a strong non-classical current component (paratropic current) that tends to create an internal magnetic field parallel to the external one. Non-aromatic systems have a vanishing net current due to similar diatropic and paratropic contributions. Although recent efforts have been made to directly measure ring currents in atoms and planar molecules[4], it is still a considerable challenge to detect them in more complex systems such as 3D clusters or nanoparticles. Their effects can be studied indirectly since they affect the shielding/de-shielding properties visible in NMR shifts of probe nuclei, which can be calculated routinely from various approaches using the density functional theory (DFT)[5,6] even for fairly large nanostructures[7–9]. However, after the pioneering work from Jusélius and colleagues[10], efficient self-consistent methods to calculate, analyse and visualize local MICs inside complex nanostructures have been lacking, preventing detailed analyses of the local electronic structure and magnetic shielding.

Ligand-stabilized metal clusters[11,12] have emerged as tunable nanomaterials with a range of potential applications in catalysis[13–16], nanomedicine[17], biological sensors[18], and $CO_2$ reduction[19,20] among others. In most cases, the metal core of these clusters is composed of noble metals, possibly doped with transition metals. The core is protected chemically by organic ligand molecules which modify its electronic structure. The electronic structure of ligand-stabilized metal clusters is commonly interpreted in terms of the superatom concept, introduced in 2008[11]. Similar to electrons residing in quantized energy shells in the ordinary atoms, metal valence electrons in superatoms can respond to an external electromagnetic field and generate MICs in these rather complicated 3D nanostructures.

Here, we have implemented a numerically efficient method to calculate gauge-including MICs in the formalism of auxiliary density functional theory. We validate the method by studying two hydrogen-containing gold-based, ligand stabilized nanoclusters $[HAu_9(PPh_3)_8]^{2+}$ (1) and $[PtHAu_8(PPh_3)_8]^+$ (2). We are able to explain the measured[21–23] anomalous difference in the proton NMR shift of the hydrogen in the metal core of these superatoms (15.4 ppm in 1 vs. 5.4 ppm in 2) by analyzing the local diatropic and paratropic MICs around the hydrogen atom. Surprisingly, Pt-doping of the gold core in (2) makes the system more aromatic, also increasing the magnetic shielding around the hydrogen. Our work presents a methodological advance for detailed analyses of the local electronic structure of the metal core and the protecting layer of superatoms, offering new possibilities to understand their growth, structure, and functions.

## Results

### Computation of the gauge-including magnetically induced current.
Historically, computations of magnetic properties led to a problem that arises from the choice of the gauge origin. In a homogeneous medium, a magnetic field B can be defined through a vector potential A(r) (B = ∇ × A(r)) but the opposite does not hold due to the arbitrary scalar function found in B = ∇ × (A(r) + ∇ Φ(r))[10]. In standard DFT computations the use of finite basis sets introduces a gauge dependence in the magnetic calculations that can be overcome only in the limit of a complete basis set. To

avoid this problem, the idea to use magnetic field-dependent basis functions goes back to the calculation of ring currents by the seminal work of London[24]. In this way the gauge dependence on the origin can be eliminated. These functions are known as gauge—including atomic orbitals (GIAO) defined in the Coulomb gauge (∇ · A(r) = 0). In large-scale computations small basis sets can be used to achieve rapid convergence[5,10].

The shielding tensor $\sigma_{C,\lambda\eta}$ for the nucleus $C$ is defined as the second derivative of the energy with respect to the external magnetic field B and the nuclear magnetic moment $\mu_C$ in the limit of zero magnetic field and zero nuclear magnetic moment ($\lambda, \eta = x, y, z$)[25].

$$\sigma_{C,\lambda\eta} = \frac{\partial^2 E}{\partial B_\lambda \partial \mu_{C,\eta}} \qquad (1)$$

An alternative expression to compute the nuclear magnetic shielding tensor via the Biot–Savart law[1,26–28] is

$$\sigma_{C,\lambda\eta} = -\frac{1}{c^2} \sum_{\alpha,\tau=x}^{z} \varepsilon_{\eta\alpha\tau} \int \frac{\alpha - C_\alpha}{|\mathbf{r} - \mathbf{C}|^3} \; \mathcal{J}_\tau^{(\lambda)}(\mathbf{r}) \; d\mathbf{r} \, , \qquad (2)$$

where $\varepsilon_{\eta\alpha\tau}(\eta, \alpha, \tau = x, y, z)$ is the Levi–Civita tensor, C is the position of the nucleus $C$ and

$$\mathcal{J}_\tau^{(\lambda)} \equiv \frac{\partial \mathcal{J}_\tau}{\partial B_\lambda} \qquad (3)$$

are the first-order elements of the tensor for the MIC density. Equating Eqs. (1) and (2) one can obtain the MIC that only includes direct dependence on basis functions, first derivatives of basis functions, density matrix, and magnetically perturbed density matrix. In this work, we take advantage of the high efficiency to calculate the nuclear shielding tensors[5] employing GIAOs to compute gauge-including magnetically induced currents (GIMIC). The implementation was done in the deMon2k[29] code (see short summary in Methods and the detailed discussion of relevant algebra in Supplementary Methods).

**Atomic and electronic structure of the studied superatoms**. We studied two previously reported phosphine-protected gold-based clusters, $[HAu_9(PPh_3)_8]^{2+}$ (1, ref. [21,30]) and $[HPtAu_8(PPh_3)_8]^+$ (2, ref. [22,23]). The position of the hydrogen atom in the metal core has been previously deduced by NMR analysis in clusters 1[21] and 2[22,23]. Their DFT-optimized structures are shown in Fig. 1. Cluster 1 shows a $C_{4v}$ symmetry in the core (formally $(HAu_9)^{2+}$) and a $C_4$ symmetry including the ligand layer. Cluster 2 shows a lower $C_2$ symmetry both in the metal core (formally $(HPtAu_8)^+$) and in the ligand layer. The two "crown-like" metal cores have an atom (Au/

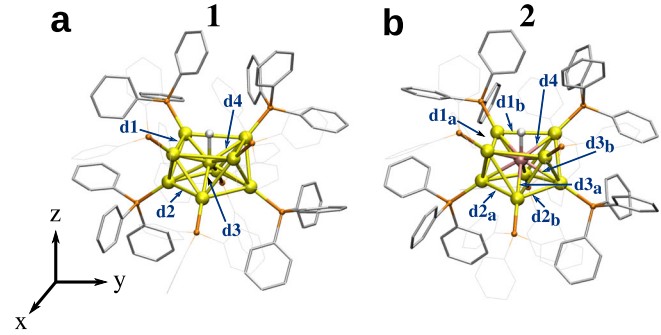

**Fig. 1 Structure of the studied clusters. a** $[HAu_9(PPh_3)_8]^{2+}$ (cluster 1 in text) and (**b**) $[HPtAu_8(PPh_3)_8]^+$ (cluster 2). Characteristic distances in metal cores are labeled (dN and dN$_{a,b}$, N = 1–4). Au: yellow, P: orange, C: grey, core-hydrogen: white. Some PPh$_3$ ligands and protons in the phenyl rings are omitted for clarity.

**Table 1 Experimental and DFT $^1$H NMR chemical shift $\delta$ and standard deviation $\sigma$ in ppm of ligand protons in clusters 1 and 2 (PBE xc functional and SDD pseudopotential).**

| | | PBE/SDD | |
|---|---|---|---|
| **1** | **Exp[a]** | $\delta$ | $\sigma$ |
| o-proton | 7.25 | 7.44 | 0.89 |
| m-proton | 6.62 | 7.27 | 0.39 |
| p-proton | 7.05 | 7.53 | 0.22 |
| **2** | **Exp[b]** | $\delta$ | $\sigma$ |
| o-proton | 7.29 | 7.68 | 1.07 |
| m-proton | 6.59 | 6.98 | 0.60 |
| p-proton | 7.02 | 7.24 | 0.22 |

[a]Reference[21]. Recorded in $CD_2Cl_2$ solution at 288 K;
[b]Reference[46]. Recorded for a $[PtAu_8(PPh_3)_8]^{2+}$ cluster in $CD_2Cl_2$ solution at 298 K.

**Table 2 Experimental and DFT $^1$H NMR chemical shift $\delta$ in ppm of the core-hydrogen in clusters 1 and 2 (PBE xc functional and SDD pseudopotential).**

| | Exp | PBE/SDD |
|---|---|---|
| **1** | 15.1[a] | 3.88 |
| **2** | 5.4[b] | −0.55 |
| $\Delta$ | 9.7 | 4.43 |

[a]Reference[21].
[b]Reference[22, 23].

Pt) at the center and can be characterized by four distances d1–d4 and their respective distortions d1$_{a,b}$–d3$_{a,b}$ (Fig. 1).

For the cluster 1 the distance d3 (2.77 Å) is longer than d4 (2.70 Å) both preserving a perfect square on top and bottom from the central Au atom (square edge being 3.57 and 3.40 Å respectively). In cluster 2 there are three characteristic distances from the central Pt atom, a shorter one to the top square (2.66 Å) and two longer (2.68 and 2.71 Å) to the bottom parallelogram. Both parallelograms are distorted from the perfect square showing longer distances on top (edges 3.65 and 3.58 Å) than on bottom (edges 3.10 and 3.15 Å). This indicates the effect of the central atom (Au/Pt) on the geometry. For cluster 1 the distance between H and the central Au is 1.72 Å and for cluster 2 the corresponding distance to central Pt atom is shorter, 1.63 Å.

The electronic structure of 1 and 2 was analyzed by projecting the frontier orbitals to cluster-centered spherical harmonics $Y_{lm}$. The projected density of states (Supplementary Note 1 and Supplementary Fig. 1) shows that both clusters are 8-electron superatoms with a closed-shell configuration and significant HOMO-LUMO energy gaps (1.88 eV for 1 and 1.93 eV for 2) as expected from their chemical formula and overall charge[11]. The Bader atomic charge analysis[31] showed for both hydrogens nearly similar, vanishing charges, −0.09 e in 1 and −0.06 e in 2. This, combined with analysis of the superatomic orbitals (Supplementary Fig. 1) indicates that both hydrogens can be thought of as neutral atoms, contributing their s-electron to the superatomic electron system. Frontier orbital analysis of 1 done in ref. [21] led to similar conclusions.

**NMR chemical shifts**. First, we calculated proton shifts for the ortho-, meta-, and para-positions in PPh$_3$ ligands for clusters 1 and 2 and compared with experimental data (Table 1 and Supplementary Table 1). This served also as an internal test for the DFT approximations (xc functional and basis set) in the calculations.

In cluster 1 the measured signals[21] fragmented into three groups. A broad signal at 7.25 ppm was assigned to the o-proton. This smeared out triplet peak was explained by the spin–spin coupling with the neighboring phosphorous and the m-proton. The 6.62 ppm triplet signal was assigned to the m-proton and the 7.05 ppm triplet was assigned to the p-proton. The DFT results showed a 7.26–7.44 ppm chemical shift for the o-proton depending on the level of theory employed. This proton showed the highest standard deviation (0.78–0.89 ppm) independently of the level of theory employed within the three o-, m-, and p-proton analyzed, in good agreement with the chemical shifts observed. Computed values for the m-proton were in the range of 7.16–7.27 ppm with standard deviation of 0.38–0.39 ppm, and for

the p-proton in the range of 7.44–7.53 ppm with standard deviation of 0.22–0.26 ppm.

Since the ligand layer of cluster 2 is similar to that of cluster 1, the measured o-, m-, and p-proton shifts showed almost identical values to the ones measured for 1. The computed values were within 0.4 ppm from the experimental values for o- and m-protons and within 0.2 ppm for p-protons (Table 1 and Supplementary Table 1). We note that all our computed values for the proton shifts are based on a single structure of the cluster and a full exploration of the distribution of the proton shifts due to ligand dynamics and potential solvent effects is not possible at the moment.

Next, we turn to the discussion of the NMR shift of the hydrogen in the metal cores of 1 and 2. The hydrogen peak of 1 was observed at 15.1 ppm[21] in comparison with the previously reported 5.4 ppm for 2[22,23] (Table 2 and Supplementary Table 2). This anomalously large difference ($\Delta = 9.7$ ppm) is rather surprising in view of similar atomic geometries of 1 and 2. Our computed values are notably shifted upfield (Table 2 and Supplementary Table 2) but show consistently a qualitatively similar difference in the range of $\Delta = 4.3–4.5$ ppm, independent of the level of DFT approximations. The systematic upfield shift and smaller $\Delta$ obtained in the computations could be explained by the limitations in the description of the electron density and the scalar-relativistic approximation used in the effective core potential[32–36]. These approximations preclude the calculations of the shielding tensor on the true all-electron density around the Pt and Au atoms including also the full spin-orbit effects.

The anomalous difference of the hydrogen shifts in 1 and 2 prompted the authors of ref. [21] to speculate the reasons arising either from different hydrogen charge or different magnetic shielding around the hydrogen in these superatoms. However, our charge analysis indicated essentially identical hydrogen charge in both systems as discussed above. Hence, we next seek the explanation by examining the computed magnetically induced currents in 1 and 2. We chose to use the PBE xc functional and the SDD pseudopotential in these computations.

**Magnetically induced currents**. The nine-component current density tensor $\mathcal{J}_\tau^{(\lambda)}$ (Eq. (3)) is a complex object and difficult to visualize. But, given a particular direction of the external magnetic field, one can analyze the current density as a vector function J(r). We chose an external magnetic field B = 1 T in a direction pointing from the central Au/Pt atom of clusters 1 and 2 to the hydrogen site and analysed J(r) in a cube containing 20 × 20 × 20 grid points centered at the central atom.

3D visualizations of J(r) (Supplementary Fig. 2 and Supplementary Movies 1 and 2) reveal a surprising qualitative indirect effect of the central Pt atom in cluster 2 on MICs around the core-hydrogen site. It is seen that Pt-doping of the gold changes significantly the diatropic current around the core-hydrogen as compared to cluster 1. To understand this interesting effect, a more quantitative analysis of this phenomenon is presented in

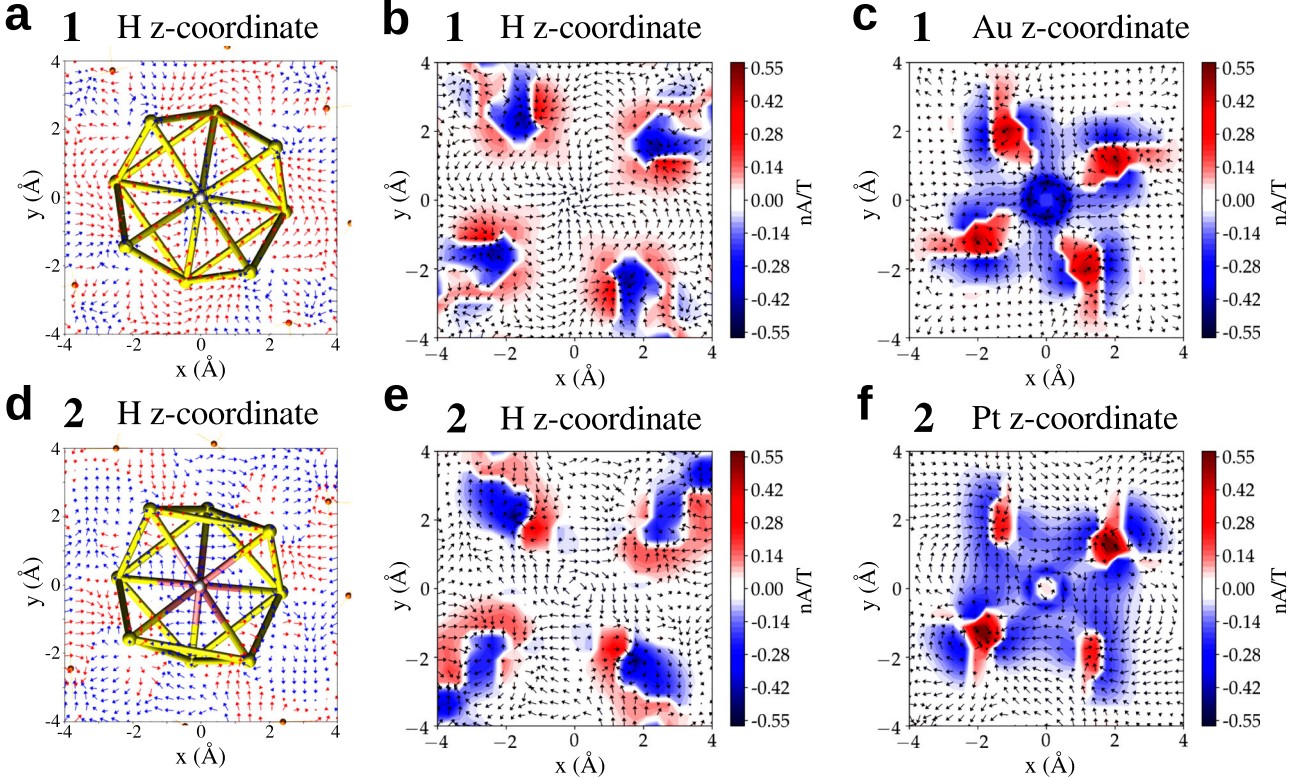

**Fig. 2 Magnetically induced currents.** Paratropic (red) and diatropic (blue) vectorial field of the magnetically induced current density J(r) for (**a–c**) [HAu$_9$(PPh$_3$)$_8$]$^{2+}$ (cluster 1) and (**d–f**) [HPtAu$_8$(PPh$_3$)$_8$]$^+$ (cluster 2). Vectors projected on an x-y plane located at the z-coordinate of the core-hydrogen and the central atom (Au/Pt) and signed modulus of J(r) are shown. The external magnetic field B is oriented perpendicular to the x–y plane and pointing towards the reader.

Figures 2 and 3 and in Table 3. Figure 2 shows the projected vectorial field of J(r) as well as the diatropic and paratropic contributions in the x-y planes containing either the core-hydrogen or the central Au/Pt atoms in 1 and 2. Pt-doping in 2 has a tiny paratropic current basically at the metal site but induces a large area of diatropic current in the metal core of 2. In cluster 1, the corresponding gold site induces a deep maximum in the diatropic current immediately around the metal site but the paratropic contribution comes into play in the vicinity, neutralizing the diatropic effect. This interplay is seen quantitatively in Fig. 3 that shows circularly integrated currents in the x-y planes containing the central metal site. Radial integrals of the contributions up to 4 Å radius show diatropic/paratropic components of (−2.619/0.715) Å$^2$nA/T in 2 as compared to (−1.626/1.184) Å$^2$nA/T in 1 (table 3). This causes the total integrated current to be −1.905 Å$^2$nA/T in 2 vs. −0.442 Å$^2$nA/T in 1 indicating an enhanced aromaticity in that plane in cluster 2.

The enhanced aromaticity in the Pt-doped cluster 2 has an indirect effect also in the plane containing the core-hydrogen, revealing a key mechanism to understand the previously reported proton NMR shifts.[21–23] Analogously to the central x-y plane discussed above, the Pt-doped cluster has also a larger diatropic environment around the core-hydrogen as compared to the all-gold cluster. Radially integrated diatropic/paratropic currents up to 4 Å in the plane containing the core-hydrogen are (−1.297/0.667) Å$^2$nA/T for 2 and (−1.034/1.253) Å$^2$nA/T for 1. As a result, the total integrated current around the core-hydrogen in 1 is *paratropic* (0.218 Å$^2$nA/T) while in 2 it is *diatropic* (−0.63 Å$^2$nA/T). This qualitative and quantitative difference in the magnetic shielding between 1 and 2 provides the explanation for the reported[21] downfield shift of the core-hydrogen in 1.

## Discussion

We have implemented and validated a numerically efficient method to calculate, analyse and visualize magnetically induced currents in nanostructures. The method is based on using gauge-including atomic orbitals in the framework of the auxiliary density functional theory. To validate the method, we studied diatropic and paratropic currents inside a metal core of two experimentally synthesized and characterized gold-based, hydrogen-containing ligand-protected clusters [HAu$_9$(PPh$_3$)$_8$]$^{2+}$ and [HPtAu$_8$(PPh$_3$)$_8$]$^+$. Our computed proton NMR shifts for the hydrogens reproduce the anomalous trend observed in the experiment, where the hydrogen in [HAu$_9$(PPh$_3$)$_8$]$^{2+}$ was significantly downfield-shifted. Analysis of the MICs in the metal cores showed that Pt doping of the gold core in [HPtAu$_8$(PPh$_3$)$_8$]$^+$, surprisingly, increases the overall contribution of diatropic currents in the core, making the cluster more aromatic. This results in a better magnetic shielding of the hydrogen in [HPtAu$_8$(PPh$_3$)$_8$]$^{2+}$ causing an upfield proton chemical shift while the net effect of the shielding currents is paratropic in a plane of the hydrogen in [HAu$_9$(PPh$_3$)$_8$]$^{2+}$ inducing de-shielding and down-field proton chemical shift.

We expect that the methodological advancement reported in this work will allow for detailed studies of local electronic structure in ligand-stabilized metal clusters which will yield an improved understanding of their physico-chemical properties and tunability for diverse applications.

## Methods

**Ground state DFT calculations.** The structures of clusters 1 and 2 were optimized using the real-space grid based GPAW[37,38] program with a uniform grid spacing of 0.2 Å. Pt, Au, C, and P atoms were described by valence of 16, 11, 4, and 5, respectively, and the frozen core approximation was used for inner electrons. The

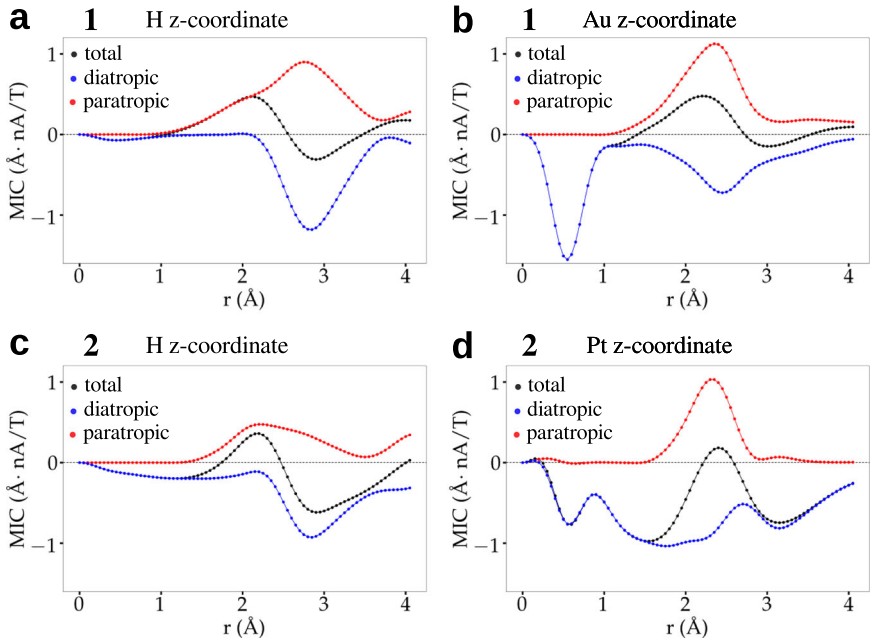

**Fig. 3 Integrated currents.** Circularly integrated current profile of the signed modulus of J(r) for (**a,b**) [HAu$_9$(PPh$_3$)$_8$]$^{2+}$ (cluster 1, top row) and (**c,d**) [HPtAu$_8$(PPh$_3$)$_8$]$^+$ (cluster 2, bottom row). The integrals are done in the x-y planes shown in Fig. 2, having either core-hydrogen or Au/Pt at the origin. Diatropic (blue), paratropic (red), and total (black) contributions are indicated.

**Table 3 Radial integrals of the circularly integrated current strength (Fig. 3) (in Å$^2$ nA/T) at H and Au or Pt x-y planes at radius r from (0,0,z) respective coordinate in clusters 1 and 2.**

| 1 | H z-coordinate | | | Au z-coordinate | | |
|---|---|---|---|---|---|---|
| r (Å) | Tot | Para. | Dia. | Tot | Para. | Dia. |
| 1 | −0.045 | 0.001 | −0.046 | −0.715 | 0.000 | −0.715 |
| 2 | 0.137 | 0.190 | −0.052 | −0.644 | 0.250 | −0.893 |
| 3 | 0.225 | 0.892 | −0.667 | −0.422 | 1.013 | −1.435 |
| 4 | 0.218 | 1.253 | −1.034 | −0.442 | 1.184 | −1.626 |

| 2 | H z-coordinate | | | Pt z-coordinate | | |
|---|---|---|---|---|---|---|
| r (Å) | Tot | Para. | Dia. | Tot | Para. | Dia. |
| 1 | −0.11 | 0 | −0.11 | −0.362 | 0.013 | −0.374 |
| 2 | −0.191 | 0.108 | −0.299 | −1.173 | 0.109 | −1.282 |
| 3 | −0.295 | 0.51 | −0.805 | −1.341 | 0.686 | −2.027 |
| 4 | −0.63 | 0.667 | −1.297 | −1.905 | 0.715 | −2.619 |

PAW setups for Pt and Au include scalar-relativistic effects. The PBE[39] functional was employed for the exchange-correlation interaction. Structure optimization was continued until the force on each atom was less than 0.05 eV/Å. To identify the superatom symmetries of the frontier orbitals, the Y$_{lm}$ analysis[11] was performed for all the clusters with a cutoff radius of 4.0 Å. A Bader charge analysis[31] was performed for the studied systems using the density obtained in a single point computation in GPAW.

**NMR chemical shifts and magnetic currents.** The magnetic shielding tensors and gauge-including magnetically induced currents were computed in deMon2k[29] code employing the Auxiliary Density Functional Theory (ADFT) approach. For the $^1$H NMR chemical shift and the MIC computations the geometries optimized in GPAW were used with PBE[39] and BP86[40,41] level for exchange-correlation effects. The Stuttgart–Dresden (SDD) pseudopotential[42] (Au and Pt are described by a valence of 19 and 18 electrons respectively) along with the DZVP[43] basis set (P, C, and H) and the effective core potentials LANL2DZ[44] along with D95[45] basis set (P, C, and H) were employed. All the computations were performed in combination with the GEN-A2* auxiliary function set. Each $^1$H chemical shift was referenced to the $^1$H TMS chemical shift computed in the respective level of theory.

## Data availability
DFT-optimized coordinates of clusters 1 and 2 are provided as Supplementary Data 1 and 2. Other data that support this study are available upon request from the authors.

## Code availability
Both DFT codes (GPAW and deMon2k) used in this work are free to download for academic use at the respective sites. GPAW can be downloaded at https://wiki.fysik.dtu.dk/gpaw/. The gauge-including MIC calculations are implemented in the version 6.1.8 of the deMon2k code (http://www.demon-software.com/public_html/index.html).

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

## Acknowledgements

This work was supported by the Academy of Finland (grants 294217, 319208, 315549), and through H.H.'s Academy Professorship. E.S. acknowledges the The Finnish Cultural Foundation for a PhD study grant. The computations were made at the Nanoscience Center of the University of Jyväskylä by utilizing the FCCI - Finnish Computing Competence Infrastructure (persistent indentifier urn:nbn:fi:research-infras-2016072533). B.Z.G. acknowledges the funding from CONACyT project CB-2015-258647.

## Author contributions

O.L.-E. conceived the concept, performed all DFT calculations, analyzed the results together with E.S. and S.M. and wrote the first manuscript draft. B.Z.-G. implemented the GIMIC method in deMon2k. H.H. supervised the work. All authors commented on the manuscript draft that was finalized by H.H.

## Competing interests

The authors declare no competing interests.
