## [Peer Review File · Nature Communications]

REVIEWER COMMENTS

Reviewer #1 (Remarks to the Author):

This manuscript reports the implementation and application of gauge-including magnetic induced current (GIMIC) calculations. The GIMIC analysis was performed for ligand-stabilized Au and AuPt 8 electron superatoms. The authors succeeded to rationalize the large difference in the chemical shift of a specific cluster hydrogen atom in these two systems based on the GIMIC. To the best of my knowledge, this is the first application of this technique to transition metal clusters. Therefore, the manuscript is novel and original. I strongly encourage the publication of this work. In general, the manuscript is well written and understandable without specific knowledge. Thus, it is well suited for a journal with broad and diverse readership. The following details the authors might want to consider for a revision:

- The tensor notation in the equations must be unified. I suggest using subscripts for the tensor component to bridge to linear algebra.
- To the best of my knowledge Eq. (2) was first suggested by Buckingham (J. Chem. Phys. 1980, 73, 5684) and more extensively used later on by the Eschrig group from Dresden.
- I think it is worth mentioning that the GIAOs are defined in the Coulomb gauge.
- From the discussion in the text, it is for me not obvious why the electron of the hydrogen contributes to the superatomic electron system.
- The authors claim that the large up-field shifts of the hydrogen chemical shifts are due to effective core potentials. I do not agree. You can find similar errors also in all-electron 3d transition metal systems. To me this looks more like an intrinsic error of current density functional approximations. They are simply not good enough for this particular property.
- In the same spirit. For the interpretation of "complex NMR spectra" the current state-of-the-art DFT shielding calculations are still not good enough.
- I really appreciate the discussion of the GIMIC in terms of diatropic and paratropic current. This is good science! However, the bridge to aromaticity, a concept based on covalent bonded organic molecules, seems to me often constructed.

Minor points:

- P3, l60: "in the limit" instead of "at the limit".
- P4, l77: "in the Method section" or similar.
- The sign ~ for indicating a range is uncommon to me.
- P6, l125 "to the discussion".
- Are there frequencies for the optimized structures? If so they should be reported.
- P10, l221 delete "and".

In conclusion, this is a pioneering work for the understanding of induced magnetic currents (and in a next step fields) in complex transition metal systems. I am convinced it will raise significant interest in the cluster community and beyond. Therefore, I strongly encourage its publication in Nat. Commun.

Reviewer #2 (Remarks to the Author):

This paper reports calculations of induced current density and chemical shifts for a pair of ligated metal clusters. The calculations are on an impressive scale. The calculations seem to me to have been correctly and competently carried out, and it is good to see systems of this size being tackled. The results give a qualitative/semi-quantitative account of the experimental chemical shifts, including a qualitative match to the literature observation of a strong effect of a metal substitution on the shift for core protons. (Direction is matched, and the magnitude of the effect is about half the experimental.) Although the abstract and text talk about explaining this observation, the explanation seems to consist of noting that the integrated property values are different because the integrand distributions are different. As a reader, I do not get the sense of why those distributions should differ, and how the results might transfer to other systems. Given the lack of generality and an incisive explanation, I do not support publication here.

Reviewer #3 (Remarks to the Author):

This manuscript presents a computational approach to calculating interatomic currents in unusual structures (i.e. a metallic nanocluster). To my reading, this computational approach aims to explain the huge difference in chemical shift of a central proton in $[\text{HAu}_9(\text{PPh}_3)_8]^{2+}$ versus in $[\text{HPtAu}_8(\text{PPh}_3)_8]^+$.

The calculation of ring current effects is of contemporary interest, and extension of the existing methods towards more complex structures is valuable.

After reading the paper a few times I have a couple of uncertainties:

* The authors calculate NMR spectra as a way to calibrate their choice of functional/basis set/pseudopotential for DFT. They report "standard deviations" for several protons, which apparently exceed the range of the calculated values (based on values given on p6). This disparity suggests that the chemical shifts of these protons are not normally distributed. If so this might bear further explanation: a standard deviation of 1 ppm (as for the o proton in 2) appears quite extreme. In any case, the ms should make clearer what the standard deviation is, how it relates to the range, and why it is so large.

* The authors present the calculated chemical shift for the internal 1H. However, their values (calculated with common pseudopotentials using different functionals) are only half of that observed by experiment. Since the chemical shift of this resonance is the prime experimental measure of the ring current (and other NMR) effects in the molecules, I am led to ask what other factors contribute to the experimental observation?

* Solvent effects?

* Is a pseudopotential insufficient?

In my view it is important to explain, or eliminate, this difference. The validity of the ring current calculations depends, in part, on the model chemistry being correct. The evidence shows that the model chemistry is unable to reproduce the key proton NMR chemical shifts for the central proton.

* The authors do not provide sufficient detail to reproduce their work. I would like to see the necessary code included in deMon2k, or made available elsewhere, prior to publication. I don't believe that "code available on request" is sufficient for a manuscript of this sort, where the code is essential to reproducing the results. Alternatively the authors must include sufficient detail to enable a reader (of this journal - Nature Comm - i.e. not necessarily an extremely specialised reader) to re-implement their algorithms in full.

* Please could the authors include the xyz coordinates as a zip file, not as text in a PDF.

I. Response to Reviewers:

Reviewer 1

This manuscript reports the implementation and application of gauge-including magnetic induced current (GIMIC) calculations. The GIMIC analysis was performed for ligand-stabilized Au and AuPt 8 electron superatoms. The authors succeeded to rationalize the large difference in the chemical shift of a specific cluster hydrogen atom in these two systems based on the GIMIC. To the best of my knowledge, this is the first application of this technique to transition metal clusters. Therefore, the manuscript is novel and original. I strongly encourage the publication of this work. In general, the manuscript is well written and understandable without specific knowledge. Thus, it is well suited for a journal with broad and diverse readership. The following details the authors might want to consider for a revision:

We thank the Reviewer for these very positive remarks.

- The tensor notation in the equations must be unified. I suggest using subscripts for the tensor component to bridge to linear algebra.

The tensor notation was unified as described in Supplementary Information, and extensive discussion on additional algebra was added to clarify the implementation.

- To the best of my knowledge Eq. (2) was first suggested by Buckingham (J. Chem. Phys. 1980, 73, 5684) and more extensively used later on by the Eschrig group from Dresden.

We thank for this comment, and have added citations to the previous work by Buckingham and Eschrig groups (references 26 - 28).

- I think it is worth mentioning that the GIAOs are defined in the Coulomb gauge.

The phrase “defined in the Coulomb gauge ($\nabla \cdot \mathbf{A}(\mathbf{r}) = 0$)” has been added (lines 63-64).

- From the discussion in the text, it is for me not obvious why the electron of the hydrogen contributes to the superatomic electron system.

To clarify this statement, a reference to the orbital analysis shown in the SI has been added to the main text (lines 103-104). Y_{lm} analysis and visualization of the frontier orbitals indicate that the clusters are 8-electron superatoms, and that the electron of the core-hydrogen is participating in HOMO states that have a clear P-symmetry about the center of mass (Supplementary Figure 1).

- The authors claim that the large up-field shifts of the hydrogen chemical shifts are due to effective core potentials. I do not agree. You can find similar errors also in all-electron 3d transition metal systems. To me this looks more like an intrinsic error of current density functional approximations. They are simply not good enough for this particular property.

We thank the Reviewer for this comment. Aside from using the pseudopotential, which are not able to capture the full electron density of the heavy metal atoms, the other main approximation is to describe the electron density of Pt and Au at the scalar-relativist level. This means that the spin-orbit effects are averaged out. We have added a sentence on lines 139-140 to this effect and

cite now also a paper by Hrobarik et al (ref. 32) that shows large upfield shifts of hydrides in transition metal / hydride complexes in case the scalar-relativistic approximation is used. Unfortunately, for these rather large metal clusters, the full relativistic calculations are not currently very feasible within the GIMIC formalism.

- In the same spirit. For the interpretation of “complex NMR spectra” the current state-of-the-art DFT shielding calculations are still not good enough.

We note that while there are the quantitative differences between the computed proton NMR shift in the core hydrogen and the experimental data, as discussed above, our main conclusions on the diatropic and paratropic currents and the (anti)aromaticity are based on analysing the systematic trends that are correctly reproduced by the theory. On the other hand, the light atoms P, C, H in the ligand layer are treated at an all-electron level (and the level of taking into account the relativistic effects is not relevant there). The computed averaged NMR shifts of o-, m-, p-protons are in a good quantitative agreement with experimental data, which is discussed below in our reply to Reviewer 3.

- I really appreciate the discussion of the GIMIC in terms of diatropic and paratropic current. This is good science! However, the bridge to aromaticity, a concept based on covalent bonded organic molecules, seems to me often constructed.

We are delighted that the Reviewer appreciates our analysis of the induced currents and our attempts to discuss the underlying physics. Aromaticity is still somewhat controversial and not well-defined physicochemical property although it is often used in computational chemistry. It is true that historically aromaticity was connected to covalent organic molecules with delocalized electrons, closed electron shells, chemical stability etc., benzene being a prototypical system. Ring currents are definitely thought to be one potential physicochemical property related to aromaticity, see e.g. the fairly recent review by M. Sola, DOI: 10.1002/wcms.1404, now cited in our paper as ref. 3. If one accepts this, then also inorganic-organic systems such the metal clusters studied in this work may exhibit aromatic properties. By the virtue of being “metallic”, those system have a finite number of delocalized electrons that can respond to an external magnetic field by induced ring-like currents. In fact, in the “superatom theory”, put forth by our group in 2008 to explain chemical stability of “magic” gold clusters stabilized by organic molecules, closed electron shells and the distinct number of delocalized electrons are fundamental concepts to classify the systems (PNAS 2008, 105, 9157). So here, we cordially disagree with the Reviewer, and like to keep the discussion connecting the paratropic/diatropic currents to (anti)/aromaticity.

Minor points: • P3, 160: “in the limit” instead of “at the limit”.

Corrected.

- P4, 177: “in the Method section” or similar.

Corrected.

- The sign for indicating a range is uncommon to me.

Corrected.

- P6, 1125 “to the discussion”.

Corrected.

- Are there frequencies for the optimized structures? If so they should be reported.

We have not calculated the frequencies for the optimized structures.

- P10, l221 delete “and”.

Corrected.

In conclusion, this is a pioneering work for the understanding of induced magnetic currents (and in a next step fields) in complex transition metal systems. I am convinced it will raise significant interest in the cluster community and beyond. Therefore, I strongly encourage its publication in Nat. Commun.

We thank the Reviewer for his/her strong support and many useful comments.

Reviewer 2

This paper reports calculations of induced current density and chemical shifts for a pair of ligated metal clusters. The calculations are on an impressive scale. The calculations seem to me to have been correctly and competently carried out, and it is good to see systems of this size being tackled. The results give a qualitative/semi-quantitative account of the experimental chemical shifts, including a qualitative match to the literature observation of a strong effect of a metal substitution on the shift for core protons. (Direction is matched, and the magnitude of the effect is about half the experimental.) Although the abstract and text talk about explaining this observation, the explanation seems to consist of noting that the integrated property values are different because the integrand distributions are different. As a reader, I do not get the sense of why those distributions should differ, and how the results might transfer to other systems. Given the lack of generality and an incisive explanation, I do not support publication here.

We thank the Reviewer for his/her critical comments. As Reviewers 1 and 3 have noted, a main contribution of this work is to present a method allowing calculations and analysis of magnetically induced currents in rather large nanostructures that are significantly more complex than any systems studied before. We chose to demonstrate the validity of the method by addressing recent unexplained experimental results regarding proton NMR shifts in a pair of ligand-stabilized gold-based nanoclusters that included a hydrogen as a dopant in the metal core. Selection of this example naturally makes the work relevant for nanocluster community, but in no way our method is restricted to work only with clusters.

The integrated distributions of the induced currents give a new way to quantitatively analyse and “classify” aromatic and anti-aromatic behaviour around a given atom in the nanosystem. Here we chose to analyse the environment around the metal-doping H atom, since the currents in that environment obviously determine the shielding / de-shielding of the proton in the NMR experiment.

Reviewer 3 :

This manuscript presents a computational approach to calculating interatomic currents in unusual structures (i.e. a metallic nanocluster). To my reading, this computational approach aims to explain the huge difference in chemical shift of a central proton in $[\text{HAu}_9(\text{PPh}_3)_8]^{2+}$ versus in $[\text{HPtAu}_8(\text{PPh}_3)_8]^+$. The calculation of ring current effects is of contemporary interest, and extension of the existing methods towards more complex structures is valuable.

We thank the Reviewer for his/her positive evaluation comments. Indeed, the main contribution of our work is to present a method to calculate and illustrate the magnetically induced currents in nanostructures and to extend the application to systems that are considerably more complex than what has been studied previously. This was also noted by Reviewer 1.

After reading the paper a few times I have a couple of uncertainties:

* The authors calculate NMR spectra as a way to calibrate their choice of functional/basis set/pseudopotential for DFT. They report "standard deviations" for several protons, which apparently exceed the range of the calculated values (based on values given on p6). This disparity suggests that the chemical shifts of these protons are not normally distributed. If so this might bear further explanation: a standard deviation of 1 ppm (as for the o proton in 2) appears quite extreme. In any case, the ms should make clearer what the standard deviation is, how it relates to the range, and why it is so large.

First, we note that our calculations of the proton NMR shifts of ligands in clusters 1 and 2 are each based on a single structure. Each cluster has 8 triphenylphosphine ligands, and the ligand layer has in total of 48 o-protons, 48 m-protons, and 24 p-protons. The average shifts for each proton class are within 0.2 to 0.6 ppm from the measured data (Table 1). The light atoms (P, C, H) are treated at all-electron level when calculating the electron density, and the rather good agreement between the averaged theoretical values and the measured ones is expected. However, as the Reviewer notes, the standard deviations particularly for o-protons in clusters 1 and 2 are rather large, 0.89 - 1.07 ppm. The smallest deviation is seen for the p-protons as 0.22 ppm both in 1 and 2. We attribute this fact to the lack of proper dynamical effects that would require long and CPU-time consuming molecular dynamics simulations. All the bonds between P atom and phenyl rings in the ligands are single bonds with low rotation barrier, consequently, it is plausible to expect that the rotations of the phenyl rings would be needed to be taken into account to produce the proper distribution and standard deviation of the proton shifts. We note that the largest discrepancy is for the o-protons that would point partially toward the metal core, partially toward the solvent side during one rotation cycle. It is therefore reasonable to expect that those protons have the largest deviation in the distribution of the shift values. Unfortunately the full exploration of the ligand dynamics is not computationally possible at the moment. A secondary, less important effect might be due to solvent-ligand interactions that are not considered in our work. We have added a sentence to this effect on p. 6 (lines 126-129): **We note that all our computed values for the proton shifts are based on a single structure of the cluster and a full exploration of the distribution of the proton shifts due to ligand dynamics and potential solvent effects is not possible at the moment.**

* The authors present the calculated chemical shift for the internal 1H. However, their values (calculated with common pseudopotentials using different functionals) are only half of that observed by experiment. Since the chemical shift of this resonance is the prime experimental mea-

sure of the ring current (and other NMR) effects in the molecules, I am led to ask what other factors contribute to the experimental observation? * Solvent effects? * Is a pseudopotential insufficient?

In my view it is important to explain, or eliminate, this difference. The validity of the ring current calculations depends, in part, on the model chemistry being correct. The evidence shows that the model chemistry is unable to reproduce the key proton NMR chemical shifts for the central proton.

As we replied to a similar comment by Reviewer 1, it is indeed plausible that the relatively large difference between calculated and measured core-hydrogen shift in both clusters 1 and 2 (Table 2) is due to (i) the pseudopotential approximation for the heavy metals Pt and Au and (ii) scalar-relativistic level of theory. For these reasons, the GIMIC calculation is not able to use the full electron density in the metal environment of the core hydrogen (an all-electron, fully relativistic description for Pt and Au would obviously fix that but it is not technically possible at the moment). However, this does not affect our main conclusions on the underlying physics producing the aromaticity/antiaromaticity that are based on the comparison of the systematic trend of the NMR shifts that our calculations correctly reproduce as compared to experimental data of clusters 1 and 2. Note also that we treat the light atoms in the ligand layer at the all-electron level, and the proton NMR shifts in ligands are much better reproduced. We have expanded the sentence on p. 7 (lines 139-140) discussing the reasons for the difference in theory and experiment, and have added a key reference that discusses the scalar-relativistic vs. fully relativistic calculations for hydride NMR shifts in transition metal-hydride complexes (ref.32).

* The authors do not provide sufficient detail to reproduce their work. I would like to see the necessary code included in deMon2k, or made available elsewhere, prior to publication. I don't believe that "code available on request" is sufficient for a manuscript of this sort, where the code is essential to reproducing the results. Alternatively the authors must include sufficient detail to enable a reader (of this journal - Nature Comm - i.e. not necessarily an extremely specialised reader) to re-implement their algorithms in full.

For the benefit of the reader, we have added extensive discussion on the mathematical details in the Supplementary Information text (pages S5-S11), to work out the theory for GIMIC calculations. The GIMIC code is now included in the master 6.1.8 version of deMon2k code and it can be downloaded from the website.

* Please could the authors include the xyz coordinates as a zip file, not as text in a PDF.

We give the coordinates now as electronic files in Supplementary Information.

REVIEWERS' COMMENTS

Reviewer #3 (Remarks to the Author):

The authors have handled all of my earlier questions and comments. I think that following the first round of reviews, the paper's content, argument, and reproducibility are all strengthened.

RESPONSE TO REVIEWERS

Reviewer #3 (Remarks to the Author):

The authors have handled all of my earlier questions and comments. I think that following the first round of reviews, the paper's content, argument, and reproducibility are all strengthened.

Response: We thank the Reviewer for this comment and look forward to the paper being published.